# Update in Diagnostics of Toscana Virus Infection in a Hyperendemic Region (Southern Spain)

**DOI:** 10.3390/v13081438

**Published:** 2021-07-23

**Authors:** Sara Sanbonmatsu-Gámez, Irene Pedrosa-Corral, José María Navarro-Marí, Mercedes Pérez-Ruiz

**Affiliations:** 1Laboratorio de Referencia de Virus de Andalucía, Servicio de Microbiología, Hospital Universitario Virgen de las Nieves, 18014 Granada, Spain; saral.sanbonmatsu.sspa@juntadeandalucia.es (S.S.-G.); irenee.pedrosa.sspa@juntadeandalucia.es (I.P.-C.); josem.navarro.sspa@juntadeandalucia.es (J.M.N.-M.); 2Instituto de Investigación Biosanitaria ibs.Granada, 18012 Granada, Spain; 3Red de Investigación Cooperativa en Enfermedades Tropicales (RICET), 28029 Madrid, Spain; 4Servicio de Microbiología, Hospital Regional Universitario de Málaga, 29010 Málaga, Spain

**Keywords:** Toscana virus, Spain, RT-PCR, automation, meningitis, encephalitis

## Abstract

The sandfly fever Toscana virus (TOSV, genus *Phlebovirus*, family *Phenuiviridae*) is endemic in Mediterranean countries. In Spain, phylogenetic studies of TOSV strains demonstrated that a genotype, different from the Italian, was circulating. This update reports 107 cases of TOSV neurological infection detected in Andalusia from 1988 to 2020, by viral culture, serology and/or RT-PCR. Most cases were located in Granada province, a hyperendemic region. TOSV neurological infection may be underdiagnosed since few laboratories include this virus in their portfolio. This work presents a reliable automated method, validated for the detection of the main viruses involved in acute meningitis and encephalitis, including the arboviruses TOSV and West Nile virus. This assay solves the need for multiple molecular platforms for different viruses and thus, improves the time to results for these syndromes, which require a rapid and efficient diagnostic approach.

## 1. Introduction

*Toscana phlebovirus* (TOSV), a species of the genus *Phlebovirus* (family *Phenuiviridae*), [1] is transmitted to humans by the bite of phlebotomine sand flies. TOSV typically causes aseptic meningitis; more rarely, cases of severe encephalitis with sequelae have been reported. It is endemic in Mediterranean countries, where seroprevalence rates are high [2,3,4,5] and most neurological infections have been documented. Beyond these borders, TOSV infections have been described in travellers returning from this area [6].

Since the first time that TOSV meningitis was documented in a Swedish tourist returning from Spain in 1991 [7], several cases have been reported in local individuals and in tourists visiting the country. The largest series of cases was first described in Granada province, located in Andalusia, a southern region [8,9]. Further studies in this area have focused on several aspects: seroprevalence surveillance [2], investigation of vectors for TOSV [2], clinical studies [2,9], diagnostic methods [10,11] and search for animal reservoirs [12].

Available data suggest that Granada province may be a hyperendemic area for TOSV for several reasons. Most human cases are concentrated in this part of the region. An overall seroprevalence rate of 25% has been found, and this percentage increases with age. A 0.05% TOSV infection rate was demonstrated in captured phlebotomine sandflies, which belonged to the species *Phlebotomus perniciosus*, *Sergentomyia minuta, P. sergenti* and *P. papatasi* [2]. Phylogenetic analysis of TOSV found in phlebotomine pools revealed the existence of a new lineage, different from the Italian strain [13]. To date, three different TOSV genotypes have been reported: genotype A (*Italian* strain), genotype B (*Spanish* strain) and genotype C, described in Croatia [14,15].

Studies in domestic animals revealed high seroprevalence rates in cats, dogs and sheep, which agrees with the breeding preferences of *P. perniciosus*, the most frequently detected phlebotomine sandfly [2,12,16,17]. Furthermore, data collected from previous studies [2,9] demonstrated that TOSV-positive phlebotomine pools were located within the same area with the highest seroprevalence rate in goats, where several human cases of meningitis had been described and where the only PCR-positive result was found in a serum sample from a goat.

Virological diagnosis can be carried out by serological studies using ELISA and/or neutralization tests [8,18]. However, in the acute phase of the infection, sensitive etiological diagnosis relies on direct detection of TOSV in the cerebrospinal fluid (CSF), either by nucleic acid amplification techniques (NAAT) or by viral culture, although the former are preferred for a rapid result, which can be achieved in few hours. Real-time NAAT have currently substituted conventional molecular methods for their undisputable advantages (reduced hands-on-time, cross contamination and time-to-results). Three real-time PCR methods have been published for the detection of TOSV RNA in clinical samples [10,19,20]. All of them have demonstrated optimal sensitivity and specificity. Recently, the combination of the three primer-and-probe sets in the same PCR demonstrated equal or improved performance compared with monoplex assays and may prevent false negative results due to mutations in genetic targets [11]. A commercial assay is available for the detection of TOSV, included in a syndromic molecular panel with other viruses [21].

However, the laboratory diagnosis of viral meningitis and encephalitis (VME) must include viruses other than TOSV, and different methods and platforms are usually used for this purpose, which may delay an early result, crucial for the management and treatment of these infections. Automated molecular assays are a good alternative to carry out VME diagnosis, but they do not include endemic viruses such as TOSV and other arboviruses, and further assays must be carried out, which makes the workflow cumbersome and laborious.

The objective of this work is to describe an update of TOSV cases in Andalusia since the first case reported in 1988, and to evaluate a new protocol designed to include TOSV molecular detection in the differential diagnosis of viral meningitis and encephalitis by an automated molecular platform.

## 2. Materials and Methods

### 2.1. Setting

The Virology Unit of the Microbiology Service, Hospital Virgen de las Nieves (VU-MS_HVN), Granada, Spain, has been the Regional Reference Laboratory for the diagnosis of VME in Andalusia since 2007. Apart from the investigation of cases from our healthcare area in Granada province, we have been receiving samples for the investigation of viruses from cases with suspicion of VME from the seven other provinces that comprise the Andalusian region. Additionally, the VU-MS_HVN participates as the reference laboratory in health programs such as the surveillance of West Nile virus and other arboviruses.

### 2.2. Portfolio for the Diagnosis of VME

Table 1 summarizes assays currently used for the diagnosis of VME in the VU-MS_HVN.

The laboratory diagnosis mostly relies on real-time RT-PCR on the cerebrospinal fluid (CSF). Serum, urine, pharyngeal exudates and stools can also be useful depending on the virus or viruses mainly associated with the clinical syndrome or the epidemiological antecedents. Serology is carried out in CSF and serum samples to investigate TOSV and West Nile virus (WNV). 

The routine molecular portfolio includes herpes simplex virus (HSV) 1 and 2, varicella zoster virus (VZV), human enterovirus (hEV), TOSV and human parechovirus (hPeV) investigation. Other viruses such as flavivirus, lymphocytic choriomeningitis virus (LCMV), JC polyomavirus (JCV), human cytomegalovirus (hCMV) and mumps virus (MuV) are investigated under specific request and/or clinical suspicion of other neurological infections and/or epidemiological background of cases or within the context of surveillance programs.

Automated real-time PCR is only implemented for HSV 1/2 detection. The remaining viruses were investigated following nucleic acids extraction with either Qiasymphony DSP Viral/Pathogen kit in a QIAsymphony SP instrument (QIagen, Hilden, Germany) or with the MagNA Pure Compact Nucleic Acid Isolation kit I in a MagNA Pure Compact instrument (Roche Diagnostics, Barcelona, Spain).

HSV 1 and 2 investigation was carried out in all CSF samples upon request. For further investigation of other viruses, leucocyte counts > 5/µL were used as the inclusion criterion. The algorithm for the molecular investigation of viruses in CSF samples is schematized in Figure 1.

### 2.3. Timeline of TOSV Investigation in Hospital Virgen de las Nieves

Two periods of TOSV investigation were differentiated, based on the laboratory methods used for TOSV investigation.

In the first period, from 1988 to 2007, viral culture of the CSF was carried out in all CSF samples, and when available, serology in the CSF and serum samples was performed. For viral culture, 200 µL of sample was inoculated in tubes with African green monkey kidney (Vero) cells. Tube cultures were incubated at 37 °C and examined daily for 14 days to observe the appearance of a cytopathic effect (CPE). Tubes with positive CPE were screened for the presence of TOSV by physicochemical assays (resistant to 5′-bromo-2′-deoxyuridine, and sensitive to chloroform and acid treatment) and confirmed by neutralization tests with specific antisera [9]. TOSV strains were retrospectively confirmed by nested RT-PCR on the cell culture supernatants [27].

In the second period, from 2007 to date, real-time RT-PCR replaced viral culture [10], which was carried out just in PCR-positive samples to recover strains for further genetic purposes.

IgG and IgM were investigated in CSF and serum samples with commercial assays following the manufacturers’ instructions (Table 1).

### 2.4. Data Recovery and Management

Laboratory results were recovered from the Laboratory Information System and an anonymous database was constructed in MS Excel 2013.

After cleaning up rough data, the results of the investigation of the following viruses were recorded: HSV 1 and 2, VZV, hEV, TOSV, WNV, MuV and LCMV.

### 2.5. Preliminary Validation on an Automated Molecular Platform for Multiplex Detection of Viruses in the CSF

In 2018, the BD MAX™ System (BDMAX) (Becton Dickinson, Madrid, Spain) was introduced in the laboratory for molecular diagnosis. It is an open platform, which performs automated nucleic acid extraction and real-time PCR directly from the clinical sample. Thus, we adapted the routine PCR assays for VME investigation to develop an automated multiplex panel in the BDMAX, using the BD™ MAX™ ExK™ TNA-3 kit. The kit allows total nucleic acid extraction and multiplexing real-time PCR assays, combining up to five fluorescence channels and two parallel reactions per sample (independently programmed if necessary) in a single cartridge, using a fixed amount of reagents and nucleic acids.

Both commercial and laboratory-developed assays (LDA) were mixed in a multiplex panel using the reagents included in the commercial assay for HSV 1/2 and LDA for VZV, hEV, TOSV and WNV (Table 1).

Each of the two multiplex PCRs of the BDMAX panel included four targets, three viral targets and an internal control. Fluorophores of the Taqman^®^ probes, master mix and amplification protocol in each multiplex PCR are shown in Table 2.

The protocol was validated with viral strains: HSV 1, HSV 2, VZV, WNV and hEV (echovirus 30) were obtained from clinical isolates and TOSV strain genotype B was obtained from the laboratory collection [13].

Ten-fold serial dilutions down to 10^−4^ of titrated strains were prepared in physiological saline sterile solution. Three aliquots of negative CSF pools were spiked with the viral suspensions and 250 µL of each sample was analyzed in parallel by the BDMAX protocol and by routine assays, using the MagNA Pure Compact as the extraction system and LC480 (Roche Diagnostics, Barcelona, Spain) for the real-time amplification. Cycle threshold (Ct) values obtained in each platform for all viruses’ amplification were recorded and used to compare the BDMAX protocol with the routine assay.

## 3. Results

### 3.1. Etiology of VME Cases in Southern Spain

Positive results of viral investigation in CSF samples from VME cases from 1988 to 2020 were recorded (Table 3). The most frequently detected viruses were hEV, TOSV, HSV 1, HSV 2 and VZV. Yearly, mean detection of all viruses increased during the second period evaluated. Overall, TOSV was the second agent detected in CSF samples, following hEV. TOSV cases were detected yearly between April and November (Figure 2).

Diagnosis of TOSV cases relied on viral culture (until 2007) and real-time RT-PCR (from 2007 to date), since serology was not always possible due to organizational issues. Only 357 serologic tests were performed in CSF or serum samples. Anti-TOSV IgM was detected in one case, which was diagnosed by RT-PCR.

It is worth noting that TOSV was the most frequently detected virus during the summer of 2010 and 2013, when the first alerts of WNV circulation in Andalusia were reported and active surveillance of WNV neurological infection was carried out. In this context, the laboratory received 39 CSF and/or serum samples for WNV investigation from suspected cases located in the affected areas, when other etiological agents were discarded in the origin laboratory. As a reference laboratory, we performed an extended molecular panel in all CSF samples, which included TOSV. Thirteen positive cases were detected, nine TOSV, two WNV, one VZV and one hEV.

Two out of the 107 TOSV cases developed a severe encephalitis with sequelae due to ischemic complications.

### 3.2. Automated Molecular Panel for the Detection of Viruses in CSF Samples

Several final concentrations of each primers/probe set for LDA were tested. The optimal performance was obtained with 0.5 µM of primers and 0.2 µM of probes for hEV, VZV and TOSV, 1 µM of primers and 0.8 µM of probes for WNV and 0.25 and 0.1 µM of primers and probe, respectively, for the RNAse P (internal control) [28].

Mean Ct values of the real-time PCR carried out in BDMAX and by the routine assay were calculated. Ct values obtained in both methods were similar and did not differ by more than 2.1 cycles (Table 4).

The total hands-on time and turnaround time for the automated BDMAX was 5 and 160 min, and for the routine assay, 30 and 135 min, respectively.

## 4. Discussion

The first description of TOSV dates back to 1971, when it was isolated from the sandfly *P. perniciosus* within an entomological surveillance in Monte Argentario (Italy) [29]. Soon after, the first cases of human TOSV neurological infections were reported in Italy, Spain, France, Greece, Cyprus and Portugal [6,30]. It has been recently suggested that the geographical distribution of TOSV may be broader than expected since other countries (Kosovo, Bulgaria, Bosnia-Herzegovina, Tunisia and Turkey) have reported TOSV cases. [15,31].

In Spain, sporadic cases and series of TOSV infections have been described in Granada, Murcia, Cataluña and Madrid [8,9,18,32,33,34].

In this work, updated data show the largest case series of TOSV neurological infection reported in the country, detected from cases from Granada and other provinces in Andalusia, southern Spain. Two plausible arguments may explain the high prevalence detected in this area: Granada province is a hyperendemic area for TOSV as it has been demonstrated in humans, phlebotomine sandflies and domestic animals [2,9,12], and routine molecular portfolio for VME diagnosis always includes TOSV.

As a regional reference laboratory, we receive samples for the investigation of suspected cases of VME and arboviral infections. It is remarkable that, apart from most cases located in Granada province, TOSV was the main virus detected in CSF samples from suspected cases of WNV neurological infection, whose samples had been sent to the reference laboratory within the active surveillance of WNV during two alerts of WNV infections in horses that occurred in west Andalusia in 2010 and 2013 [35,36]. Moreover, during an outbreak of 70 WNV fever cases in Andalusia in 2020 [37], TOSV was also detected in one case by RT-PCR, since an extended molecular portfolio was applied to all CSF samples. The implementation of reliable RT-PCR has probably helped in improving TOSV detection, since only 17 cases had been reported until 2004 [9], mostly detected by viral culture. In 2007, real-time RT-PCR was introduced to the portfolio of the laboratory [10], and up to now, 90 more cases have been detected.

The outcome was favorable in most cases. Only two cases developed severe encephalitis with sequelae, corresponding to individuals with underlying immunosuppressive conditions [38].

Nine out of the 17 first TOSV cases (53%) were detected in August until 2004 [9], whereas in the next years, TOSV cases were widely distributed from July to October. Although this finding can be interpreted as being because of climatic change, the improvement in laboratory diagnosis by the introduction of efficient real-time PCR in 2007 may also have contributed to the increase in the detection period.

Thus, TOSV infections are probably underestimated in Spain. The inclusion of this viral target in syndromic panels is desirable in order to assess the real role of TOSV in VME.

VME are the main causes of acute neurological infections [39]. The reference methods to diagnose VME are nucleic acid amplification techniques. In most situations, a clinical laboratory may need different platforms and assays to cover the portfolio of the most prevalent viruses. Arboviruses must be investigated in endemic areas, where neurological infections by these agents can be frequent. There are few commercial assays for multiplex detection of viruses in CFS samples that include TOSV [21], and these methods lack automation, which should be a premise for a rapid result.

Although this work shows a preliminary evaluation, it demonstrates that the automated method described here is an open and versatile system. It can be adapted for multiplex PCRs that include the most prevalent viruses in each geographical area, and avoids the use of different panels for the complete coverage of the virological investigation.

## Figures and Tables

**Figure 1 viruses-13-01438-f001:**
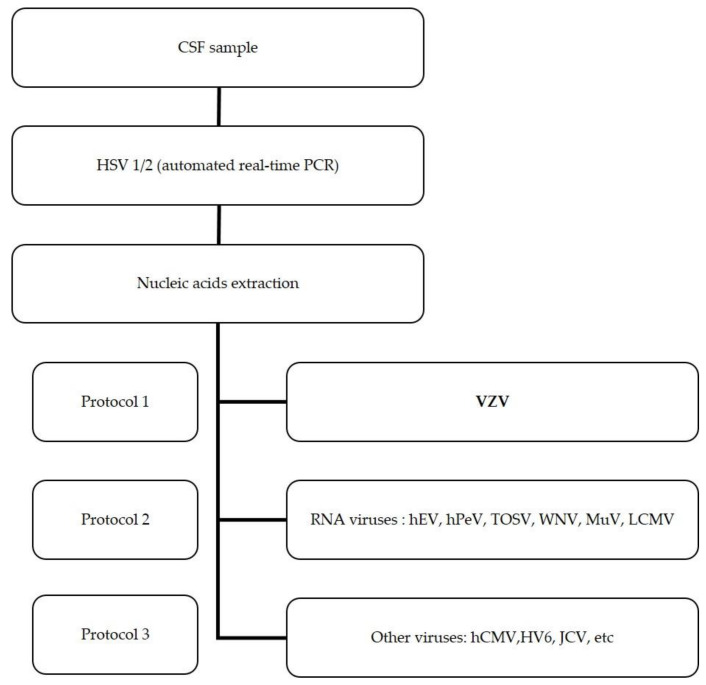
Algorithm of the molecular detection of viruses involved in acute meningitis and encephalitis. Protocols 1–3 were simultaneously performed from the nucleic acids extract after HSV 1 and were discarded. Real-time PCR and RT-PCR were carried out for the detection of DNA and RNA viruses, respectively.

**Figure 2 viruses-13-01438-f002:**
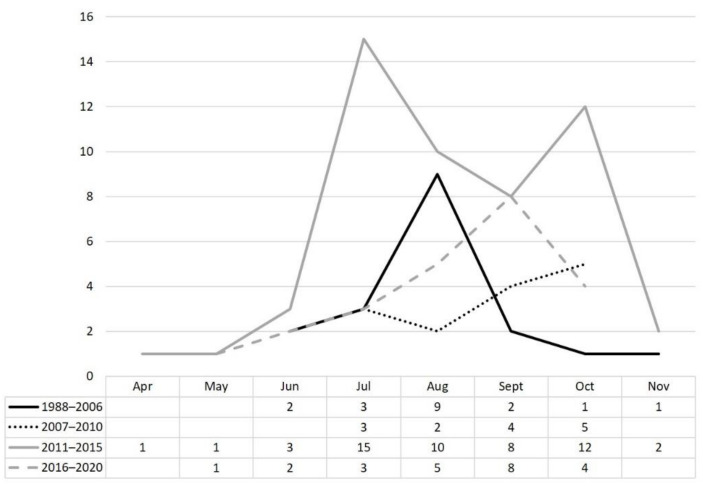
Temporal distribution of TOSV cases.

**Table 1 viruses-13-01438-t001:** Assays for the detection of VME in Hospital Virgen de las Nieves.

Viral Target	Commercial Assay	Laboratory Developed Real-Time Assay (Gene Target) [Reference]
HSV 1 and 2	Simplexa™ HSV 1 and 2 Direct Kit (DiaSorin; Saluggia, VC, Italy)	-
hEV	Xpert^®^ EV (Cepheid, Barcelona, Spain)	5′ UTR [22]
VZV	RS-VZV (AB analitica, Padova, Italy)	ORF 29 [Laboratory design]
TOSV	TOSCANA VIRUS IgG-IgM (Sandfly Fever Virus) (Diesse, Siena, Italy) IIFT: Sandfly fever virus mosaic 1 (IgM) (Euroimmun, Diagnostics, Madrid, Spain)	S segment [10]
MuV	-	F gene [23]
LCMV	Indirect immufluorescence	S segment [Laboratory design]
hCMV	kPCR PLX CMV DNA assay (Siemens, Erlangen, Germany)	-
hPeV	-	5′ NC [24]
Flavivirus	-	NS5 [25]
JC polyomavirus	kPCR PLX JC DNA assay (Siemens)	-
WNV	WNV IgM and IgG ELISA (Euroimmun)	3′ UTR [26]

**Table 2 viruses-13-01438-t002:** Characteristics of the multiplex molecular panel validated in the BDMax system.

PCR Tube	Target	5′-Fluorophore of the Taqman^®^ Probe	Master Mix	Amplification Protocol
1	HSV1	TexasRed	Simplexa HSV 1/2 Direct (DiaSorin, Madrid, Spain)	97 °C/5 min + 45 ciclos: 97 °C /10 s + 60 °C/35 s
	HSV 2	6-FAM		
	VZV	HEX		
	IC (A/B) *	Quasar 670 / Cy5		
2	hEV	6-FAM	qScript XLT 1-step RT-PCR (QuantaBio, VWR, Llinars del Vallés, Spain)	50 °C/10 min + 98 °C/1 min + 45 ciclos: 98 °C/10 s + 50.3 °C/30 s + 60 °C/30 s
	TOSV	HEX		
	WNV	TexasRed		
	IC B	Cy5		

* A, exogenous internal control included in the commercial assay for HSV ½; B, endogenous internal control (RNAse P).

**Table 3 viruses-13-01438-t003:** Etiology of VME cases, 1988–2020.

Virus	n	%
Human enterovirus	616	61.5
Toscana virus	107	10.7
Varicela zoster virus	97	9.7
Herpes simplex virus 1	60	6
Herpes simplex virus 2	18	1.8
Mumps virus	22	2.2
JC polyomavirus	4	0.4
Lymphocytic choriomeningitis virus	2	0.2
West Nile virus	76	7.6
Total	1002	

**Table 4 viruses-13-01438-t004:** Ct values obtained with the BDMAX and routine assay for each viral strain.

Virus/Dilution	Mean Ct Value (BDMAX /Routine Assay)					ΔCt *
	-	10^−1^	10^−2^	10^−3^	10^−4^	
HSV 1	12.7/13.1	14.8/16.2	18.8/19.5	21.3/22.2	26.3/26.7	−1.4, −0.4
HSV 2	12.5/14.1	16.2/18.9	19.8/21.6	23/25	26.3/28.2	−2.7, −1.6
VZV	18.1/16.2	21.4/20	24.4/24.4	28.3/27.7	33/31.5	0, +1.9
hEV	12.8/14.7	17.9/18.1	20.1/21.4	23.3/24.9	27.4/27.1	−1.9, +0.3
TOSV	14.5/12.4	17.9/16.1	21.3/20	24.1/23.8	26/25.9	+0.3, +2.1
WNV	19.2/17.5	23/20.9	25.8/23.9	28.8/26.2	31/29.8	+1.2, +2.1

* Maximum, minimum difference between Ct in BDMAX and Ct in the routine assay.

## Data Availability

Not applicable.

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
