# Peer review of "Update in Diagnostics of Toscana Virus Infection in a Hyperendemic Region (Southern Spain)"

_viruses, 2021, doi:10.3390/v13081438_

Round 1

Reviewer 1 Report

In the manuscript, the authors overview laboratory-confirmed Toscana phlebovirus (TOSV) infections at a single centre in an endemic region, within a significantly-broad timespan of over 30 years. They further report a preliminary evaluation of an automated commercial nucleic acid testing platform, adapted to accomodate multiple viral agents of central nervous system infections including TOSV.  In general, TOSV is a widespread but frequently neglected and underdiagnosed agent of neuroinvasive infections around Mediterranean countries. Therefore, the manuscript will add significantly to the currently-available information on TOSV and provide insights from the diagnostic perspective.  However, the impact of the study will surely increase if the authors can provide further information/data on the following issues:

- (Table 1) does the table provide data from PCR-positive cases only? any concominant serology results available?  Given the significantly-long duration of specimen acquisition, more information other diagnostic assays, would be interesting. 

- One cannot help but wonder about clinical presentation and follow-up in diagnosed individuals. A brief information on symptoms, routine CSF findings and outcome would significantly enhance the impact of the paper. 

- (Table 4, on the comparison of the findigns by the routine and adapted platforms) any statistical evaluation of two approaches, repeated testing etc.?

- (page 5 lines 158-159) what was the frequency of TOSV detection? would a figure providing annual detection frequencies make sense?

Minor issues:

- (lines 19-20) “the assay solvents”? typing error?

- (page 2 lines 60-61) available or published?

- (page 4 line 128) please provide company information as elsewhere in the manuscript

- (page 5 line 143) which strains were used for validation?

- (page 5 lines 154-155) “second period” meaning 2007 and onward?

- (page 7 line 206) (Fernandez-Roldan) personal communication?

- (page 7 line 218) please delete “of” and rephrase.

Author Response

ANSWER TO REVIEWER 1 COMMENTS:

Major points:

- (Table 1) does the table provide data from PCR-positive cases only? any concominant serology results available?  Given the significantly-long duration of specimen acquisition, more information other diagnostic assays, would be interesting.

Does the reviewer refer to the first table of the results section (table 3 in the revised version)? It’s true that the table is not correct as it reflects etiology of VME cases by direct assays, i.e. viral culture and PCR for most virus and by PCR and serology for WNV. However, we give the whole data and prefer focusing on TOSV.

In the case of TOSV, we have only provided positive cases obtained by viral culture and PCR since serologic testing has been erratic in our laboratory. We only have data from 357 serologic tests, most of them carried out in CSF and some of them in serum samples (not always available). Moreover, only one IgM positive was detected in this series.

To solve the reviewer’s comment, we have included a brief explanation to this issue following the table.

- One cannot help but wonder about clinical presentation and follow-up in diagnosed individuals. A brief information on symptoms, routine CSF findings and outcome would significantly enhance the impact of the paper. 

We agree that this information may improve the impact. However, we don´t have information on symptoms of all cases. Available data for us include the diagnostic algorithm and the outcome. Thus, some additional information has been added in the Material & Methods and Results section, explaining the inclusion criteria the laboratory follows for viruses’ investigation and the main findings related to the patients’ outcome, respectively. Moreover, a brief description on the favourable outcomes of all except two cases is included in the results and discussed (previously published data; reference included in this revised version).

- (Table 4, on the comparison of the findigns by the routine and adapted platforms) any statistical evaluation of two approaches, repeated testing etc.?

We agree with the reviewer that a statistical analysis would be valuable. However, we have carried out a preliminary evaluation, only 3 replicates of each dilution and virus were tested in independent runs. Thus, a statistical analysis may be biased due to the sample count. Further evaluation would be necessary and is currently carried out, mainly forced by the WNV surveillance program.

We have stated that this is a preliminary evaluation in the corresponding parts of the text.

- (page 5 lines 158-159) what was the frequency of TOSV detection? would a figure providing annual detection frequencies make sense?

We have included in the new version a figure showing the temporal yearly detection of TOSV in CSF, grouped in 4 periods. Certainly, we agree that these data provide valuable and clearer information.

Minor errors:

  • (lines 19-20) “the assay solvents”? typing error?: corrected
  • (page 2 lines 60-61) available or published?: it’s “published”, corrected
  • (page 4 line 128) please provide company information as elsewhere in the manuscript: this information has been included throughout the manuscript.
  • (page 5 line 143) which strains were used for validation?: a text, explaining strains used for the validation has been added.
  • (page 5 lines 154-155) “second period” meaning 2007 and onward?: subsection 2.3 has been modified to clarify the laboratory methods carried out by period, which may help the reader in understanding this part of the results.
  • (page 7 line 206) (Fernandez-Roldan) personal communication?: The appropriate reference is cited now.
  • (page 7 line 218) please delete “of” and rephrase: deleted.

Reviewer 2 Report

It is an interesting revision article about Toscana Virus transmitted by sand flies in Spain, endemic region.

TITLE: The word epidemiology should be deleted from title as nothing has been said on this subject.

INTRODUCTION:

line 49 there is a typo. Change "sand fly" by "sandlfy"

line 54 there is a typo. Change "been" by "be"

line 56, please put the full text in brackets here because it is the first time you mention the acronym CSF.

line 67, please put the full text in brackets here because it is the first time you mention the acronym VME.

MATERIALS AND METHODS:

line 86, please put the full text in brackets here because it is the first time you mention the acronym VNW.

Figure 1: After nucleic acids extraction of the remaining virus, do you do the diagnosis using  PCR or RT- PCR? Please specify

Line 114-115: Were viral culture of the CSF and serology of the CSF and serum samples carried out from 1988 until 2007? Please specify

line 146, please put the full text in brackets here because it is the first time you mention the acronym BDMAX.

line 148, please put the full text in brackets here because it is the first time you mention the acronym Ct.

Section 3.2 should be in Materials and Methods section

DISCUSION:

 line 212-213 there is a typo in the citation (Navarro 2008, revisión 212 eimc). Please correct.

In general the discussion is weak and can be improved. For example in the Results section you stated "All TOSV cases were yearly detected between May and October" but you didn´t discuss it

Author Response

ANSWER TO REVIEWER 2 COMMENTS

TITLE: The word epidemiology should be deleted from title as nothing has been said on this subject: done

INTRODUCTION:

  • line 49 there is a typo. Change "sand fly" by "sandlfy": corrected here and elsewhere.
  • line 54 there is a typo. Change "been" by "be": DONE
  • line 56, please put the full text in brackets here because it is the first time you mention the acronym CSF: done
  • line 67, please put the full text in brackets here because it is the first time you mention the acronym VME: done

MATERIALS AND METHODS:

  • line 86, please put the full text in brackets here because it is the first time you mention the acronym VNW: done
  • Figure 1: After nucleic acids extraction of the remaining virus, do you do the diagnosis using PCR or RT- PCR? Please specify: a note has been added to the figure legend to clarify this point

Line 114-115: Were viral culture of the CSF and serology of the CSF and serum samples carried out from 1988 until 2007? Please specify: subsection 2.3 has been modified to clarify the laboratory methods carried out by period.

line 146, please put the full text in brackets here because it is the first time you mention the acronym BDMAX: done

line 148, please put the full text in brackets here because it is the first time you mention the acronym Ct: done

Section 3.2 should be in Materials and Methods section: the first part of this section has moved to the Material and Methods section, and the text has been adapted to this modification (tables re-numbered). The second part should be kept in the Results, as it shows the final conditions adopted for the method.

DISCUSION:

 line 212-213 there is a typo in the citation (Navarro 2008, revisión 212 eimc). Please correct: done (the reference is cited and included in its corresponding section (last reference).

In general the discussion is weak and can be improved. For example in the Results section you stated "All TOSV cases were yearly detected between May and October" but you didn´t discuss it.

We have provided additional data in the results section, which may clarify the results, and changed some parts of the discussion accordingly.

Round 2

Reviewer 1 Report

The manuscript revision has been carried out as suggested. This reviewer detected a typo (page 5 line 141: preliminary), which can be fixed during prrof correction.